



# Technical note: Euclidean Distance Score (EDS) for algorithm performance assessment in aquatic remote sensing

Amanda de Liz Arcari[1], Juliana Tavora[2], Daphne van der Wal[1,3], and Mhd. Suhyb Salama[1]

[1]Department of Water Resources, Faculty of Geo-Information Science and Earth Observation (ITC), University of Twente, Enschede, OV, 7522 NH, Netherlands
[2]IFREMER, DYNECO, Hydrosedimentary Dynamics Laboratory (DHYSED), 29280 Plouzané, France
[3]Department of Estuarine and Delta Systems, NIOZ Royal Netherlands Institute for Sea Research, Yerseke, ZE, 4401 NT, Netherlands

**Correspondence:** Amanda de Liz Arcari (a.delizarcari@utwente.nl)

**Abstract.** In the absence of community consensus, there remains a gap in standardized, consistent performance assessment of remote-sensing algorithms for water-quality retrieval. Although the use of multiple metrics is common, whether reported individually or combined into scoring systems, approaches are often constrained by statistical limitations, redundancy, and dataset- and context-dependent normalizations, leading to subjective or inconsistent interpretations. To address this, we propose the Euclidean Distance Score (EDS), which integrates five statistically appropriate and complementary metrics into a composite score. Capturing three core aspects of performance (regression fit, retrieval error, and robustness), EDS is computed as the Euclidean distance from an idealized point of perfect performance, providing a standardized and interpretable measure. We demonstrate the applicability of EDS in three scenarios: assessing a single algorithm for different retrieved variables, comparing two algorithms on shared retrievals, and evaluating performance across contrasting trophic conditions. By offering an objective framework, EDS supports consistent validation of aquatic remote sensing algorithms and transparent comparisons in varied contexts.

## 1 Introduction

Assessing the quality of remote sensing products is vital to monitoring and understanding aquatic ecosystems. The growing availability of algorithms that estimate water quality variables from optical signals, along with their varying transferability across regions and conditions, underscores the need for rigorous performance metrics to evaluate them (Werdell et al., 2018). In the absence of consensus in the community, studies often report multiple metrics (e.g., Pearson's $r$, $r^2$, slope, mean absolute error, mean percentage error, bias, root-mean-squared error) to quantify agreement between retrievals and observations. While reporting several metrics is important to capture complementary aspects of performance (Stow et al., 2009; IOCCG, 2019), some are partially redundant in what they represent, and not all are well-suited for the statistical properties of bio-optical data (Werdell et al., 2018). This can introduce subjectivity and risk biased or inconsistent assessments (Simão et al., 2024; Seegers et al., 2018).



Bio-optical data (i.e., optical proxies of biogeochemical state such as absorption, scattering, attenuation coefficients, and constituent concentrations) typically follow log-normal distributions and span wide dynamic ranges (Campbell, 1995). Consequently, several common error metrics are less appropriate. For example, root-mean-squared error is most informative when
residuals are approximately Gaussian and homoscedastic. Because it squares residuals, it disproportionately penalizes large deviations and becomes sensitive to skew and outliers (Morley et al., 2018). Therefore, for this type of data, simple deviation metrics are preferred over sums of squares (Seegers et al., 2018), and calculations on log-transformed values are recommended (IOCCG, 2019; EUMETSAT, 2021; Mélin and Franz, 2014). Additional good practices include using medians to reduce outlier influence, ensuring symmetric treatment of over- and under-prediction, and favoring relative errors to maintain
meaning across large ranges (O'Shea et al., 2023; Pahlevan et al., 2022).

To mitigate subjectivity from multiple standalone metrics, scoring-based and comparative approaches have been explored. Brewin et al. (2015) introduced a composite score aggregating multiple metrics, later adapted by Neil et al. (2019). However, potential redundancy among metrics may bias comparisons (Seegers et al., 2018). Seegers et al. (2018) also proposed a pairwise comparison metric (% wins), identifying the algorithm with the lowest residual error per observation. The method also
accounts for algorithm robustness, as failed retrievals penalize the score. While valuable for ranking multiple algorithms, these comparative methods are less suited for evaluating a single algorithm across variables, environmental conditions, or datasets.

Graphical multidimensional tools are also used to summarize performance across metrics. Taylor diagrams have supported cross-dataset comparisons in aquatic remote sensing (Arabi et al., 2020), but are limited to a fixed set of metrics and lack scoring capability. Radar/star plots offer more flexibility and are increasingly used to derive composite scores: Tran et al. (2023) defined
a summary indicator based on the area enclosed by polygons linking normalized metrics; Simão et al. (2024) incorporated the distance from the polygon centroid to the center to reward both high and balanced performance; Subirade et al. (2024) summed normalized scores to rank algorithms. These enhance within-study interpretation, but reliance on normalization constrains broader applicability: scores become tied to dataset-specific ranges. Shifts in value distributions or context (e.g., different optical water types/trophic states, oceanic vs. coastal) change the normalization baseline, and heterogeneity in variables and
units further limits comparability. As a result, such indices remain inherently relative and are not comparable across broader contexts.

Considering these challenges, we propose the Euclidean Distance Score (EDS), designed to provide a standardized and broadly applicable framework for evaluating aquatic remote sensing algorithms. EDS quantifies performance as the Euclidean distance from an idealized result, integrating complementary metrics selected for their suitability to bio-optical data. By avoid-
ing redundancy and dataset-specific normalization, EDS supports consistent evaluation of water quality retrievals and strengthens the reliability of remote sensing products that support aquatic biogeochemistry and ecosystem monitoring. The following sections review Euclidean distance in composite scoring (2), detail metric selection (3), introduce the EDS formulation (2), and demonstrate its application (5).





## 2 Euclidean Distance in Composite Scores

The Euclidean distance quantifies the separation between two points in $n$-dimensional space. For points $\mathbf{x} = (x_1, \ldots, x_n)$ and $\mathbf{y} = (y_1, \ldots, y_n)$, it is defined as:

$$d(\mathbf{x}, \mathbf{y}) = \sqrt{\sum_{i=1}^{n} (x_i - y_i)^2}. \tag{1}$$

This provides a straightforward and interpretable measure of dissimilarity and forms the basis for composite performance scores across scientific domains. When multiple performance indicators are treated as spatial dimensions, a single score can be 60 derived from their distance to an idealized reference point.

In hydrology, the widely used Kling-Gupta Efficiency (KGE) (Gupta et al., 2009) combines correlation, bias, and variability into one score, computed as the Euclidean distance from a perfect model. Similarly, Hu et al. (2019) introduced the DISO metric, which integrates correlation, absolute error, and RMSE to evaluate climate model performance (Hu et al., 2019). These examples demonstrate the usefulness of Euclidean distance in synthesizing multiple indicators into a robust, interpretable score.

Developing effective composite scores requires careful selection of individual components. Chosen metrics should be relevant to the application domain and reflect distinct performance aspects to avoid redundancy. While equal weighting supports balance, formulations could also be adapted to emphasize specific aspects based on application needs. Additionally, limiting the number of metrics helps preserve robustness, as high dimensionality can reduce the reliability of distance-based scores (Samantaray et al., 2024). The next section outlines the rationale for selecting the individual performance metrics to be used in our 70 composite approach.

## 3 Selection of Individual Metrics

For the construction of our composite score, we define three core dimensions of retrieval performance: (A) regression fit, which reflects how well the estimated values follow the pattern of the observed data; (B) retrieval error, which quantifies the magnitude and direction of deviations between estimates and observations; and (C) retrieval robustness, which describes the algorithm's 75 ability to produce valid outputs consistently. For each dimension, we selected metrics suited to characterizing performance in the context of bio-optical data.

### 3.1 Regression Fit

To characterize the regression fit between estimated and observed values, we employ a Type II regression (reduced major axis) to account for uncertainty in both in-situ and retrieved values. Calculations are performed in logarithmic space to accommodate 80 the log-normal distribution of bio-optical data (Campbell, 1995). The following metrics are selected:

- **Pearson correlation coefficient** ($r$)**:** Quantifies the strength and consistency of the linear association between log-transformed estimated and observed values.



– **Slope** ($m$)**:** Describes the proportionality of estimates relative to observations, indicating whether the dynamic range is well represented.

## 3.2 Retrieval Error

To characterize retrieval error, we selected metrics that quantify the typical magnitude of deviation and the systematic bias between estimated and observed values. We adopt two metrics proposed by Morley et al. (2018) to address limitations of traditional measures such as root-mean-squared error and mean average percentage error (Salama et al., 2022), which are advised for variables that span over orders of magnitude. These have been increasingly adopted in recent aquatic remote sensing studies (O'Reilly and Werdell, 2019; O'Shea et al., 2021, 2023; Pahlevan et al., 2020). Advantages of these metrics include their relative (percentage) format for interpretability and comparability, symmetric treatment of over- and under-estimations, and robustness to outliers and skewed distributions (Morley et al., 2018).

Both metrics are based on the accuracy ratio ($Q_i$), defined as:

$$Q_i = \frac{E_i}{O_i} \tag{2}$$

where $E_i$ and $O_i$ are the estimated and observed values, respectively.

Based on the accuracy ratio, the retrieval error metrics are calculated as follows:

– **Median Symmetric Accuracy** ($\epsilon$)**:** Measures the typical relative error between estimates and observations. It is defined as:

$$\epsilon = 10^{\mathrm{median}_{i=1,\ldots,n}(|\log_{10}(Q_i)|)} - 1 \tag{3}$$

– **Signed Symmetric Bias** ($\beta$)**:** Quantifies the systematic tendency of the algorithm to overestimate or underestimate observations. The median of the logarithmic accuracy ratio is first defined as:

$$M = \mathrm{median}_{i=1,\ldots,n}(\log_{10}(Q_i)) \tag{4}$$

Using this definition, $\beta$ is computed as:

$$\beta = \mathrm{sign}(M) \times (10^{|M|} - 1) \tag{5}$$

It should be noted that although Morley et al. (2018) formulated these metrics using the natural logarithm, the choice of logarithm base is arbitrary. Furthermore, since $\log(Q) = \log(E) - \log(O)$, these metrics are conceptually analogous to the mean absolute error and bias calculations as recommended by Seegers et al. (2018). The differences are the use of median instead of mean aggregation (as the distribution of $\log(Q)$ may not be symmetric (Morley et al., 2018)), and that 1 is directly subtracted to rescale the value into a percentage deviation term.





 ## 3.3 Retrieval Robustness Metrics

Retrieval robustness characterizes the algorithm's ability to consistently produce valid outputs across the dataset. In real-world remote sensing applications, retrieval algorithms may fail to converge or generate valid estimates under challenging observational conditions. To quantify this aspect of performance, we use:

– **Valid Retrieval Ratio ($n$):**

$$n = \frac{n_{\text{valid}}}{n_{\text{total}}} \tag{6}$$

where $n_{\text{valid}}$ is the number of retrievals yielding valid outputs and $n_{\text{total}}$ is the total number of retrieval attempts.

## 4 The Proposed Euclidean Distance Score (EDS)

The Euclidean Distance Score (EDS) aggregates the selected individual metrics into a single composite score that quantifies the overall retrieval performance by calculating the Euclidean distance from an idealized result. The score is computed as:

$$\text{EDS} = 1 - \sqrt{(r-1)^2 + (m-1)^2 + \epsilon^2 + \beta^2 + (n-1)^2} \tag{7}$$

In this formulation, an ideal algorithm would achieve $r = 1$, $m = 1$, $\epsilon = 0$, $\beta = 0$, and $n = 1$, yielding an EDS of 1. As deviations from these values increase, the EDS decreases proportionally, reflecting poorer algorithm performance.

To help interpret the EDS, benchmarks can be set based on typical thresholds for acceptable retrieval performance. For example, with $\epsilon$ and $\beta$ at 30% (commonly accepted error in aquatic remote sensing (IOCCG, 2019)), $r = 0.5$, slope = 1.2 (or 0.8), and $n = 0.9$, the EDS is approximately 0.31. This suggests that results exceeding this benchmark could be considered generically indicative of good retrieval performance in this context.

EDS can be extended to evaluate overall performance when algorithms retrieve multiple variables by averaging individual EDS values into an aggregated score. This approach enables flexible comparisons depending on evaluation priorities: if one variable is of primary interest, the highest individual EDS can guide selection; if balanced performance is desired, the aggregated EDS identifies the most consistent algorithm across variables. Additionally, weights can be applied to reflect variable importance, allowing tailored assessments to specific application needs.

## 4.1 Relative influence of performance components

The design of the EDS balances three aspects of retrieval performance: regression fit, retrieval error, and retrieval robustness. When errors ($\epsilon$ and $\beta$) are below 1 (i.e., within 100%), the slope ($m$) is within $\pm 1$ of the ideal (i.e., $0 \leq m \leq 2$), and the correlation is positive ($r \geq 0$), the maximum contribution to the distance calculation is 2 for regression fit (if $r = 0$ and $m = 0$ or $m = 2$), 2 for retrieval error (if $\epsilon = 1$ and $\beta = 1$), and 1 for retrieval robustness (if $n = 0$). This means that regression fit and retrieval error typically weigh equally and collectively more than retrieval robustness, prioritizing output quality while





still accounting for algorithmic stability. However, if errors or slope deviations are excessive, or correlation is negative, these factors can contribute disproportionately to the distance calculation, strongly penalizing the final EDS.

### 4.2 Metrics dependency

The EDS is designed to capture complementary aspects of retrieval performance while minimizing metric redundancy. Although the selected metrics are conceptually independent, correlations may still arise in practice due to shared responses to specific data patterns or algorithm behavior. For example, $\epsilon$ and $\beta$ may co-vary when there is consistent over- or underestimation, and in some datasets, $r$ and $m$ may show strong empirical correlation. However, in reduced major axis regression, $m$ is not directly dependent on $r$, as it is governed by the ratio of standard deviations and only adopts the sign of $r$ (Smith, 2009). Such empirical correlations do not indicate conceptual redundancy, as each metric captures a distinct and relevant aspect of retrieval performance.

## 5 Application Examples

We demonstrate the EDS by evaluating bio-optical retrievals against in situ reference data, focusing on inherent optical properties (IOPs) commonly used to characterize water quality.

The analysis illustrates three main use cases of the EDS framework. First, we evaluate the retrieval of different variables using a single algorithm, including three IOPs—phytoplankton absorption ($a_\phi(443)$), detrital and gelbstoff absorption ($a_{\mathrm{dg}}(443)$), and particulate backscattering ($b_{\mathrm{bp}}(443)$)—and the diffuse attenuation coefficient ($K_d(489)$) (a quasi-IOP (Yu et al., 2016)). Second, we compare the performance of two algorithms retrieving the same variables. Finally, we examine how performance for $K_d(489)$ varies across water types, defined by chlorophyll-$a$ thresholds (0.1 and 1 mg·m$^{-3}$) for oligotrophic, mesotrophic, and eutrophic conditions.

### 5.1 Dataset and Processing

The dataset used is the NASA bio-Optical Marine Algorithm Dataset (NOMAD) (Werdell and Bailey, 2005), a publicly available, globally distributed collection of 4,459 in situ bio-optical observations across diverse aquatic environments. Each record includes spectral upwelling radiance ($L_w$) and downwelling irradiance ($E_s$) at 21 nominal wavelengths (411–683 nm), alongside coincident optical property measurements and metadata.

Remote sensing reflectance ($R_{rs}$) was calculated as $L_w/E_s$. The IOPs of interest—$a_\phi(443)$, $a_{\mathrm{dg}}(443)$, and $b_{\mathrm{bp}}(443)$—were derived using standard relationships. Solar geometry was computed from coordinates and time, with a fixed sensor viewing geometry (40° zenith, 135° azimuth relative to the sun (Mobley, 1999)).

$R_{rs}$ spectra and geometry serve as input to 2SeaColor algorithm (Salama and Verhoef, 2015), to retrieve the three IOPs and $K_d(489)$, and to the Quasi-Analytical Algorithm (QAA) (Lee et al., 2002), to retrieve only the IOPs.

Post-processing involved filtering retrievals with variable-specific thresholds to exclude unrealistic values and define invalid retrievals:



- $a_\phi(443)$: 0.0001–11 m$^{-1}$

- $a_{\rm dg}(443)$: 0.0001–9 m$^{-1}$

- $b_{\rm bp}(443)$: 0.000215–12 m$^{-1}$

- $k_d(489)$: 0.001–10 m$^{-1}$

Minimum values for $a_\phi$ and $a_{\rm dg}$ reflect WET-Labs ac9 instrument uncertainty (Brewin et al., 2015), while the lower bound for $b_{\rm bp}$ corresponds to the minimum seawater backscattering at 443 nm, estimated from dataset temperature and salinity. Upper

limits are based on extreme values reported in the literature (Jorge et al., 2021).

Finally, the EDS is computed from valid retrievals, integrating the individual metrics described in Section II.

## 5.2 Use Case 1: Evaluating Variable-Specific Retrieval Performance

The first use case demonstrates how EDS captures performance differences across multiple retrieval targets using 2SeaColor (Fig. 1). Among the evaluated variables, $K_d(489)$ achieved the highest EDS (0.80), reflecting excellent overall performance

with perfect convergence (3346/3346), low errors, and strong correlation. In contrast, $b_{\rm bp}(443)$ shows the lowest EDS (0.06), despite low bias ($\beta = -4\%$) and acceptable median symmetric accuracy ($\epsilon = 24\%$). The poor score is mostly driven by an exaggerated slope (1.86). Intermediate scores for $a_\phi(443)$ (0.45) and $a_{\rm dg}(443)$ (0.25) reflect strong correlation but higher errors.

## 5.3 Use Case 2: Comparing Algorithms

This second use case illustrates how EDS supports direct comparison of algorithm performance for the same variables. Fig. 2 shows QAA retrievals for $a_\phi(443)$, $a_{\rm dg}(443)$, and $b_{\rm bp}(443)$, while 2SeaColor results are in Fig. 1. For $a_\phi(443)$, both algorithms achieve the same EDS (0.45), despite differences in individual metrics. QAA performs better for $a_{\rm dg}(443)$ (EDS = 0.41 vs. 0.25), with stronger correlation and better alignment with the 1:1 line. For $b_{\rm bp}(443)$, both perform poorly, but QAA yields worse results due to a steeper slope and stronger negative bias ($\beta = -37\%$ vs. $-4\%$).

Aggregated EDS scores are nearly identical (0.25 for 2SeaColor and 0.24 for QAA) indicating comparable overall performance. However, if one variable is particularly important for a given application, the algorithm with the highest EDS for that variable should be preferred.

## 5.4 Use Case 3: Evaluating Retrieval Performance Across Water Types

The third use case illustrates how EDS enables performance evaluation across water types with varying dynamic ranges. Fig. 3

shows $K_d(489)$ results grouped by trophic state: oligotrophic, mesotrophic, and eutrophic. In oligotrophic waters, EDS is 0.50 despite the lowest correlation ($r = 0.55$), driven by low errors. Mesotrophic waters show the highest overall performance (EDS = 0.76), with a balanced metric profile. Eutrophic waters also yield high performance (EDS = 0.73), with the highest $r$ due to greater variability, though increased scatter is reflected in a higher $\epsilon$.





**Figure 1.** Scatterplots of in situ vs. retrieved values using 2SeaColor, with annotations for the correlation coefficient ($r$), regression slope ($m$), median symmetric accuracy ($\epsilon$), signed symmetric bias ($\beta$), valid retrieval ratio ($n$, valid retrievals/all input points), and the Euclidean Distance Score (EDS). Red contours indicate data density.



**Figure 2.** Scatterplots of in situ vs. retrieved values using QAA. Panel annotations follow the same convention as in Figure 1.

## 6   Conclusions

The Euclidean Distance Score (EDS) provides an objective framework for synthesizing algorithm performance in aquatic remote sensing. By integrating regression fit, retrieval error, and robustness into a single, interpretable score, we show how EDS





**Figure 3.** Scatterplots of in situ vs. retrieved $K_d(489)$ for three water types: oligotrophic, mesotrophic, and eutrophic. Panel annotations follow the same convention as in Figure 1.

enables consistent comparisons across variables, algorithms and environmental contexts. Its transparent formulation based on established metrics supports reproducibility, while its simplicity facilitates broader adoption in both research and operational



contexts. In addition, the framework can be adapted to emphasize specific aspects of performance when required, ensuring
flexibility across diverse applications. Although EDS offers a unified view of performance and supports decision-making,
reporting individual metrics remains essential for diagnosing specific retrieval limitations and guiding targeted improvements
of algorithms.

*Data availability.*  The bio-Optical Marine Algorithm Dataset (NOMAD) used in this study is publicly available from NASA's Ocean Biology
Processing Group at: https://seabass.gsfc.nasa.gov/wiki/NOMAD

*Author contributions.*  Amanda de Liz Arcari: conceptualization, methodology, software, formal analysis, data curation, visualization, and
writing – original draft. Juliana Tavora: validation and writing – review and editing. Daphne van der Wal: supervision and writing – review
and editing. Mhd. Suhyb Salama: conceptualization, methodology, supervision, funding acquisition, and writing – review and editing.

*Competing interests.*  The authors declare that they have no conflict of interest.

*Acknowledgements.*  The authors thank the NASA Ocean Biology Processing Group and the individual data contributors for compiling,
maintaining, and providing public access to the SeaWiFS Bio-optical Archive and Storage System (SeaBASS), and in particular the NASA
bio-Optical Marine Algorithm Dataset (NOMAD). Cruise-level details and contributor information are available through the SeaBASS
database (https://seabass.gsfc.nasa.gov/wiki/NOMAD).



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
