# Peer review of "Technical note: Euclidean Distance Score (EDS) for algorithm performance assessment in aquatic remote sensing"

_EGUsphere, 2025_

## Author Comment (AC1)

**Authors Reply to RC1: 'Comment on egusphere-2025-4343', Richard Stumpf, 20 Oct 2025**

*Reviewer comments appear in **bold**. Authors' responses are in plain text and indented for clarity.

**The paper proposes a strategy for algorithm comparison/evaluation by designing a single metric to combine multiple metrics. This is a solid progression from previous work (referenced) that looked at metrics for algorithm assessment. The "Euclidan Distance Score" (EDS) is a strong approach to summarize the data. A critical objective of the authors is to identify only the metrics that are relevant, and summarize those, rather than to include lots of (often closely related) metrics and leave it to the reader to make sense of them. I will say that this paper was a pleasure to review, and it will become an excellent paper that should be quite important (and hopefully well used). But it does need revision to make sure it is correct.**

**A concern with comparing metrics is how to "normalize" those metrics that have quite disparate ranges. This approach addresses it by treating ratios & proportions, and so are unitless. That provides a good approach that is not arbitrary. While it does not force results to be between 0 and 1, it is set up with two strong conditions. An EDS = 1 is "perfect". Any EDS < 0 is unacceptably poor, and each of the input parameters to the EDS are typically going to be between 0 and 1. The ones that are not (proportional slope deviation, proportional error, and proportional bias), are really unacceptable if the values exceed 1.**

**I have two large concerns that should be directly solvable. First: the parameters to input. Second is whether the configuration of the equation parameters is correct.**

> We thank the reviewer for the encouraging evaluation of our proposed method and for the constructive assessment and suggestions. The identification of the key issues is appreciated. We address each group of comments in detail below.

**The inputs are R (Pearson correlation coefficient), linear regression slope calculated in log space (m), median ratio error (e ~ epsilon), Median ratio bias (B ~ beta), and valid retrieval ratio (n).**

**The question is: are these all robust and independent? Of these, e, B, and n are quite good. It is true that e and B are not actually independent, but as there appears to be no robust means of separating the two (de-biasing the error means calculating mean errors, rather than median errors, which gets into non-robust methods), so we will go with it.**

**As a practical matter a competent product should tend toward a bias ratio of 1. If it does not, then it is punished relatively severely, as e >= B. A biased "low error" model will probably do worse than an unbiased relatively high error model. This should be noted in the paper.**

> We agree that, in practice, a good retrieval result should tend toward negligible systematic bias ($\beta \approx 0$; note that $\beta$ is already rescaled to deviation instead of ratio in our proposed formulation. More details on how we make that more clear in answer below). In the EDS framework, the typical magnitude error ($\epsilon$) and the systematic bias ($\beta$) are both derived from the same accuracy ratio $Q = E/O$ and, by construction, $| \beta |\leq \epsilon$, with equality occurring when deviations are purely systematic.

This dependence implies that bias-driven degradations tend to reduce EDS through two pathways. For example, adding a constant offset of $+0.05$ to all log accuracy ratios in a tight, unbiased distribution:

$$\log_{10}(Q_i) = [-0.03, -0.01, 0, 0.01, 0.03],$$

yields:

$$\log_{10}(Q_i)' = [0.02, 0.04, 0.05, 0.06, 0.08],$$

for which $\beta$ increases from 0 to approximately 12% and $\epsilon$ increases from approximately 2% to 12%. Assuming a fixed $n = 0.98$, the corresponding EDS decreases from 0.97 to 0.83.

In contrast, increasing magnitude error through additional scatter without introducing bias, for example:

$$\log_{10}(Q_i)' = [-0.08, -0.06, 0, 0.06, 0.08],$$

increases $\epsilon$ from 2% to approximately 15% while $\beta$ remains zero, resulting in a smaller EDS decrease from 0.97 to 0.85. This illustrates that systematic bias can penalize EDS more strongly than a comparable increase in magnitude error from unbiased scatter, consistent with the reviewer's observation.

We note, however, that this behaviour does not imply that EDS systematically favours unbiased retrievals. For instance, at fixed $n = 0.98$, a case with $\epsilon = \beta = 10\%$ yields a higher EDS (0.86) than a case with $\epsilon = 20\%$ and $\beta = 0\%$ (EDS = 0.80), demonstrating that total error magnitude dominates the score in this case.

We will include this discussion in the revised manuscript, alongside a thorough sensitivity analysis (more details on Reply to Reviewer 3).

**At lines 24-28 the paper notes the problem of using root-mean-square error metrics. This is a critical point. Basically, the paper sets out that robust metrics should be used, which is why the paper proposed median e and B. However, Pearson regression and linear regression slope are least squares solutions. Thiel-Sen slope, or an equivalent, should be used for the slope. This is necessary, as many optical models (or for that matter, many models) often deviate at very low or very high values. That statistical leverage will severely alter a least squares regression slope, but not a robust slope metric.**

**Regression as a metric has an additional critical flaw: it normalizes to the standard deviation of the data. Therefore, an exact subset of a population that has a smaller range will have a lower R value than the population. (Worse, as observed in Seegers et al., a low error method with a small range of data will have a lower R values than a higher error method with a much larger range of data.) This problem is also seen in Figure 3. Oligotrophic water has the smallest error, but a low R value. The problem is the narrow range of data. Conversely if the range is large enough, R provides no useful information, both good and poor models can have high R values. Because of this problem, including R means that EDS values are not be comparable across the different data sets. (There is a good discussion of the problem of R by a top statistician https://www.stat.cmu.edu/~cshalizi/mreg/15/lectures/10/lecture-10.pdf ) .**

**By the way, R and linear regression slope are not independent, slope = S_y/S_x * R.**

**As to the input metrics, based on appropriate and consistently robust metrics, the appropriate ones would then appear to be**

**1 median (Thiel=Sen) slope, to capture whether the data generally behaves well across the range. (I will say that I don't really like slope, but I do not see a better option, as that would involve more complex partitioning alternatives that are difficult to standardize.)**

**2 median error**

**3 median bias**

**4 retrievals n.**

We thank the reviewer for raising this important discussion and for directing us to relevant literature. We acknowledge the concerns raised regarding regression-based diagnostics, in particular their sensitivity to data range and leverage effects, which can compromise comparability across datasets.

These considerations directly motivated the revisions to the EDS framework. In the revised formulation, neither regression slope nor correlation coefficient are retained as components of the composite score. We adopted a conservative approach and restricted the EDS to three metrics: a robust measure of error magnitude ($\epsilon$), a robust measure of systematic bias ($\beta$), and a term capturing retrieval feasibility ($n$).

While robust slope estimators such as the Theil–Sen method mitigate sensitivity to outliers and leverage, slope estimates remain inherently dependent on the range of the evaluated data regardless of the regression method employed. When values span a narrow range (e.g., within a single optical water type), slope estimates become poorly conditioned and associated with increased uncertainty, such that small data perturbations can lead to large variations in the estimated slope. Consequently, slopes are not always reliable as performance metrics or directly comparable across datasets or stratifications. In addition, our redundancy analysis (see reply to reviewer 3) indicated that including any slope-based diagnostic introduces overlap with bias-related metrics.

By excluding regression-based diagnostics and focusing on robust, range-independent measures, the revised EDS avoids the limitations highlighted by the reviewer and improves comparability across datasets and stratifications. We have also expanded the discussion about the rationale for metric selection and this will be included the revised manuscript.

**Median error and bias do not appear to be correctly specified in EDS equation (7). As these are ratios, shouldn't they be $(e - 1)^2$ and $(B-1)^2$? Both are defined as a ratio of E/O (expected/observed), so a value of 1, is perfect, and should reduce to zero. Equation would be:**

**EDS = 1 – sqrt [ $(m-1)^2$ + $(e-1)^2$ + $(B-1)^2$ + $(n-1)^2$ ]**

As defined in Equations (3) and (5), $\epsilon$ and $\beta$ are expressed as proportional deviations from unity (ratio minus one) and are therefore zero at perfect agreement. This formulation allows zero-centred deviations to be combined directly in Equation (7) without additional transformation.

The ratio-minus-one form was chosen to preserve interpretability: for example, a value of 0.10 directly indicates a 10% deviation, avoiding potentially confusing statements such as "an error of 1.1 corresponds to a 10% deviation."

To be as clear as possible, we will make it more explicit in the revised manuscript that $\epsilon$ and $\beta$ are defined as zero-centred proportional deviations. The proposed description of the metrics to be included is the following:

"*Building on the concept of the accuracy ratio introduced by Tofallis (2015), Morley et al. (2018) proposed a set of metrics designed for variables that span several orders of magnitude. These metrics are based on the logarithm of the accuracy ratio:*

$$log(Q_i) = log\left(\frac{E_i}{O_i}\right)$$

*To quantify typical error magnitude while ensuring symmetry between over- and under-estimation, the absolute value of the logarithmic accuracy ratio is considered. Interchanging estimated and observed values therefore yields the same error magnitude. These values are aggregated across all estimation–observation pairs using the median, providing robustness to skewed distributions and outliers:*

$$M = median(|log_{10}(Q_i)|)$$

*The aggregated value is exponentiated to return to multiplicative space and shifted relative to the ideal ratio of unity by subtracting one, yielding the Median Symmetric Accuracy ($\epsilon$):*

$$\epsilon = 10^M - 1$$

*This formulation produces an unsigned, **zero-centred measure of typical proportional deviation from perfect agreement**, directly interpretable as a fractional (or percentage) error.*

*Using the same underlying quantity, systematic bias is quantified by taking the median of the signed logarithmic accuracy ratio:*

$$M' = median(log_{10}(Q_i))$$

*And defining the Symmetric Signed Percentage Bias ($\beta$) as:*

$$\beta = sign(M)\left(10^{|M'|} - 1\right)$$

*where the sign indicates systematic over- or under-estimation and the magnitude reflects the **typical proportional bias relative to the ideal value of zero**.*"

**The authors might ponder thought experiments as examples (suggestion only). I did only one. An algorithm that has all results on an exact line with a slope of 1, but is severely biased. Error (e ~ epsilon) and (B ~ beta) will be equal. If the bias is 2x, which is a low performance, the EDS would return a value of zero.**

We thank the reviewer for this constructive suggestion. We agree that thought experiments can be valuable for illustrating the behaviour and interpretation of the EDS.

To support this, we have generated a geometric representation of the revised EDS in a three-dimensional $(\beta, \epsilon, n)$ space (Figure 1), following the exclusion of regression slope and Pearson correlation coefficient from the score. This representation explicitly shows the domain of admissible metric combinations and how the EDS varies as a function of error magnitude, systematic bias, and retrieval robustness.

The reviewer's example of an algorithm producing estimates that lie on an exact line with slope one but are severely biased corresponds to the case $\epsilon = |\beta| = 1$ i.e. a systematic multiplicative bias of a factor of two. For this configuration, considering $n = 1$, the revised EDS would be approximately −0.41. This value reflects a retrieval that is strongly inaccurate and can also be used to motivate practical benchmark value, for instance illustrating that an EDS around −0.4 corresponds to a consistently biased retrieval with errors on the order of 100%, which would generally be regarded as poor performance in practice.

In the revised manuscript, we will explicitly include such thought experiments derived from the geometric representation to illustrate how different performance regimes map onto the EDS.

[Figure]

Figure 1. Geometric representation of the Euclidean Distance Score (EDS) in the three-dimensional $(\beta, \epsilon, n)$ space. The ideal retrieval corresponds to $(\beta, \epsilon, n) = (0,0,1)$. The shown domain is restricted to metric combinations satisfying $|\beta| \le \epsilon$, consistent with their definition. For visualization purposes, EDS values are displayed over the range $[-2,1]$.

---

## Author Comment (AC2)

**Authors Reply to RC2: 'Comment on egusphere-2025-4343', Anonymous Referee #2, 19 Nov 2025**

*Reviewer comments appear in **bold**. Authors' responses are in plain text and indented for clarity.*

**The manuscript addresses an issue of great importance, which is the assessment and comparison of remote sensing algorithms, given the non-normal distributions of most bio-optical variables. Generally, the text is brief, and the points made by the authors are clear and relevant. I have recognized the need for a robust assessment method for some time, and I see the advantages of the proposed one. However, I think the manuscript is too brief at times, and some sections may benefit from more in-depth explanation.**

> We thank you for the encouraging and constructive review of our paper. Your suggestions are much appreciated. We address each in detail below.

**Firstly, regarding the assumptions made, the reduced major axis regression is mentioned several times, but I find that additional information is needed to clarify the significance of the problem. To illustrate my point, I found an interesting publication by Bilal et al. (2022) in the Encyclopedia of Mathematical Geosciences (https://doi.org/10.1007/978-3-030-26050-7_270-1). This work discusses the presence of errors in both the dependent and independent variables in geosciences, which is exactly what I find missing in this text to highlight the value of this study.**

> We thank the reviewer for this suggestion and for providing a helpful reference. We agree that this context was not sufficiently articulated in the original manuscript.

> In the revised manuscript, we will add a dedicated section discussing several commonly used metrics in the field, with their characteristics and limitations, including regression-based diagnostics:

> "*Regression parameters are commonly reported to assess the proportionality and offset between estimated and observed values. The slope is more widely reported to evaluate performance, as it describes how variability in the observations is scaled by the estimates, indicating whether the dynamic range is compressed or expanded. The intercept represents an additive offset at the reference origin.*

> *Several regression approaches can be used to estimate these parameters. Ordinary least squares (OLS, type-1 regression) minimizes vertical residuals and implicitly assumes that the observed values are error-free. To account for uncertainty in both estimated and observed quantities, type-2 regressions such as reduced major axis (RMA) are frequently employed in aquatic remote-sensing studies. RMA minimizes the perpendicular distance to the regression line and is symmetric, such that the slope of $E$ (estimated) regressed on $O$ (observed) is the inverse of $O$ regressed on $E$. Nevertheless, both type-1 and type-2 regressions are least-squares methods and therefore remain sensitive to the statistical distribution of the data, outliers and leverage points. More robust alternatives, including Theil–Sen (asymmetric) and Passing–Bablok (symmetric) regression methods, reduce sensitivity to outliers by relying on median-based estimators.*

*Nevertheless, regardless of the method employed, regression parameters are sensitive to the range of the evaluated data. When values span a narrow range (e.g. within a single optical water type), slope estimates become poorly conditioned and associated with increased uncertainty, such that small perturbations in the data can lead to large variations in the estimated slope. As a result, regression parameters are not always reliable and directly comparable across datasets or stratifications."*

We note, however, that considering the mentioned limitations and identified redundancy with bias metrics (see Reply to Reviewer 3 for more details), regression slope (and Pearson correlation coefficient, more details on the following answer) is no longer retained a component of the composite EDS. This revision removes reliance on regression assumptions in the score formulation itself, but we suggest that regression diagnostics are still reported, where relevant, as descriptive information.

**Furthermore, I am curious as to why the Pearson correlation coefficient was selected instead of the Mann-Kendall test, which does not have such strict assumptions, particularly when not all variables have ideal log-normal distributions and log-transformation does not always ensure normality.**

We thank the reviewer for this question and for highlighting the limitations of Pearson correlation in the presence of non-normal and heteroscedastic data.

In the revised manuscript, we will add a dedicated section discussing several commonly used metrics in the field, including the characteristics and limitations of Pearson r:

*"The Pearson correlation coefficient ($r$) and the coefficient of determination ($r^2$) are widely used metrics to represent the goodness of fit between estimated and observed values. While these metrics are useful for characterizing linear association, they do not quantify agreement or accuracy per se (Bland & Altman, 1986). In particular, they are insensitive to systematic bias: an algorithm may consistently over- or under-estimate observations while still yielding a high correlation coefficient if the relative ordering of values is preserved. Furthermore, both $r$ and $r^2$ are sensitive to outliers and strongly dependent on the dynamic range of the data. When values span a wide range, high correlation coefficients may be obtained even in the presence of substantial absolute or relative errors, whereas restricted ranges can yield low correlation despite good agreement."*

Considering these limitations (some of which also apply to other rank-based association measures, such as the Mann–Kendall statistic), and in light of the redundancy identified between correlation and error-magnitude metrics, the correlation coefficient is no longer retained as a component of the EDS in the revised formulation.

**Secondly, a paper of this nature, aiming to establish a certain assessment standard, should provide a broader explanation of the somewhat arbitrary nature of the logarithm selection mentioned in line 106. I remember being quite confused about this when I was a beginning researcher, and I believe that a methods paper should explain it more thoroughly. Similarly, the**

**definition of the number of valid retrievals in Equation 6 seems rather vague. I would expect a more specific definition of what "valid" means here and how it may affect the results.**

We thank the reviewer for this comment and agree that these points require clearer explanation in a methods-oriented paper.

Regarding the logarithm selection, we will expand the revised manuscript to explicitly clarify that the choice of logarithm base is mathematically arbitrary:

"*It should be noted that although Morley et al. (2018) formulated these metrics using the natural logarithm, the authors note that the choice of logarithm base is arbitrary, provided that the corresponding antilogarithm is applied consistently. In practice, using different logarithm bases does not change the meaning of the metric, as long as the same base is used throughout the calculation.*"

We will also expand the definition of the valid retrieval ratio to more clearly specify what constitutes a "valid" retrieval:

"*The valid retrieval ratio (n) is a measure of retrieval robustness. It is defined as the fraction of cases for which an algorithm produces a valid estimate relative to the number of available reference observations. A valid retrieval is defined as one for which the algorithm converges and returns physically plausible parameter values within predefined bounds. These bounds are established a priori based on instrument uncertainty, known physical limits, and extreme values reported in the literature. Retrievals falling outside these bounds are considered non-valid and excluded from the valid retrieval count.*

*The number of valid retrievals is commonly reported in the aquatic remote-sensing literature, but is often treated as a descriptive statistic rather than being explicitly incorporated into the quantitative evaluation or ranking of algorithm performance. Recognizing the importance of retrieval feasibility, particularly for inversion-based algorithms that may fail to converge or produce non-physical solutions, some studies have started explicitly integrated measures of retrieval success into their scoring or ranking frameworks (Brewin et al., 2015; Müller et al., 2015; Seegers et al., 2018).*"

**Lastly, I appreciate presenting real-life examples. However, I believe that adding a few more commonly used metrics, such as the root mean squared error, and discussing their limitations could help illustrate why the proposed approach is more robust.**

We thank the reviewer for this suggestion and agree that including commonly used metrics can help contextualize the proposed approach. In the revised manuscript, we will expand the example section to report additional traditional metrics alongside the EDS results and to explicitly discuss their behaviour, limitations and how they compare.

As mentioned in the previous answers, we have developed a dedicated section discussing commonly used metrics in the field, with their characteristics and limitations (a summary table is shown in the Reply to Reviewer 4). This will provide a good basis for discussion when we present the examples.

**To summarize, I find this work to be much needed and valuable, and it is already well-written. However, to convince sceptics and encourage broader application, I recommend providing additional explanations for those entering the field who may not understand the jargon or have not yet grasped all the challenges related to assessing optical algorithms.**

We thank the reviewer for this encouraging assessment. We believe the suggested additions will help broaden the applicability and understanding of the proposed approach.

---

## Author Comment (AC3)

**Authors Reply to RC3: 'Comment on egusphere-2025-4343', Anonymous Referee #3, 21 Nov 2025**

*To improve readability the responses below address the reviewer's points directly; the full reviewer comments are not repeated.

We thank the reviewer for this detailed critique. Many of the concerns raised, particularly those related to metric redundancy, dominance of individual components, and interpretation of example cases, highlight limitations of our initial formulation and manuscript. In response to these comments (and related feedback from other reviewers), we have substantially revised the EDS framework to strengthen its conceptual and statistical basis. We also acknowledge that some statements in the original manuscript were imprecise or insufficiently supported by quantitative analysis. These will be revised and clarified accordingly in the updated manuscript.

Below, we provide detailed responses to the reviewer's comments. Related concerns have been grouped and addressed together for clarity, as outlined below.

| Author's Reply | Reviewer comments |
|---|---|
| A. Normalization, Sensitivity and relative contributions | 1 (partially), 4, 7, 8 |
| B. Misinterpretation of Reduced Major Axis (RMA) Regression | 2 |
| C. Ideal point | 3 |
| D. Redundancy | 1 (partially), 5 |
| E. Behaviour in the Example Cases | 6 |
| F – Treatment of Variables | 9 |

**A. Normalization, Sensitivity and Relative Contributions**

The reviewer is correct in noting that a composite, distance-based score requires careful consideration of the relative scaling, influence and sensitivity of its components. We also agree that our earlier statement that the components "typically weigh equally" was imprecise and will be revised in the manuscript. In response to these concerns, we revised the EDS formulation and explicitly examined how the remaining metrics contribute to the distance calculation both in theory and in practical cases.

Figure 1 illustrates a geometric representation of the revised EDS in a three-dimensional space, following the exclusion of regression slope and Pearson correlation coefficient from the score (see Reply D for details). Of the three remaining metrics, the valid retrieval ratio ($n$) is naturally bounded between 0 and 1 and referenced to an ideal value of unity. Median Symmetric Accuracy ($\epsilon$) and Symmetric Signed Percentage Bias ($\beta$) are dimensionless, defined relative to an ideal value of zero, with magnitudes that directly reflect fractional deviations from perfect agreement. Although they are formally unbounded and may exceed unity, this property is retained so that extreme deviations are strongly penalized rather than compressed through imposed bounds. For retrievals with errors and biases below 100% (corresponding to reasonably performing retrievals in practice), all metrics are of order unity and therefore contribute comparably to the distance. This is reflected in the

approximately isotropic geometry of the high-EDS region, indicating that no single metric is implicitly favoured in that regime. In contrast, values of $\epsilon$ or $|\beta|$ exceeding unity correspond to strongly degraded retrievals and dominate the distance, displacing the solution away from the ideal point in the EDS space.

[Figure]

*Figure 1. Geometric representation of the Euclidean Distance Score (EDS) in the three-dimensional $(\beta, \epsilon, n)$ space. The ideal retrieval corresponds to $(\beta, \epsilon, n) = (0,0,1)$. The shown domain is restricted to metric combinations satisfying $|\beta| \leq \epsilon$, consistent with their definition. For visualization purposes, EDS values are displayed over the range $[-2,1]$.*

Nevertheless, having comparable numerical scales does not imply that all metrics exert equal influence on the EDS across its admissible domain. As mentioned by the reviewer, the components exhibit different empirical variances: the agreement-based terms ($\epsilon$ and $\beta$) may span a wide range depending on retrieval quality, whereas the valid retrieval ratio ($n$) is bounded and, in most realistic applications, concentrated near its ideal value of unity. To quantify how these differences translate into effective influence on the score, we performed a pointwise sensitivity analysis based on the analytical gradient of the Euclidean distance, identifying the locally dominant direction of score variation at each admissible point.

The resulting dominance structure, illustrated in Figure 2, shows that variations in error magnitude ($\epsilon$) control the sensitivity of the score over most of the admissible space. Systematic bias ($\beta$) does not emerge as a dominant sensitivity on its own, but attains equal influence with $\epsilon$ along a narrow, well-defined surface where $|\beta| = \epsilon$ and both exceed $n - 1$. Sensitivity to the valid retrieval ratio ($n$) is comparatively smaller over large portions of the space, but becomes dominant where $\epsilon$ and $\beta$ are low.

When averaged over the explored domain (restricted to EDS > −2), the mean relative sensitivities are 1.106 for $\epsilon$, 0.338 for $\beta$, and 0.116 for $n$. These values describe the average local responsiveness of the EDS to perturbations in each component across the admissible space. Stratifying the analysis by EDS (Table 1) reveals a transition in sensitivity regimes: near-optimal retrievals (EDS > 0.75) are most sensitive to $n$, whereas increasingly degraded retrievals exhibit progressively stronger sensitivity to $\epsilon$. The comparatively lower sensitivity associated with $\beta$ does not imply a negligible contribution of bias to the distance. Rather, it reflects the constraint $|\beta| \leq \epsilon$, which limits the independent variability of bias and confines its influence on the distance to specific regions of the space where systematic over- or underestimation occurs.

[Figure]

Figure 2. Sensitivity dominance structure of the Euclidean Distance Score (EDS) in the three-dimensional ($\beta, \epsilon, n$) space under the constraint $|\beta| \leq \epsilon$. Colored regions indicate the metric to which the EDS is locally most sensitive, based on the analytical gradient of the distance

*Table 1. Mean relative sensitivities of the Euclidean Distance Score (EDS) with respect to error magnitude ($\epsilon$), systematic bias ($\beta$), and valid retrieval ratio (n), computed over non-overlapping EDS bins. Sensitivities quantify the average local response of the score to perturbations in each component.*

| EDS range | $\langle Sn \rangle$ | $\langle S\beta \rangle$ | $\langle S\epsilon \rangle$ | Dominant sensitivity |
|---|---|---|---|---|
| EDS > 0.75 | 0.425 | 0.023 | 0.103 | n |
| 0.50 < EDS ≤ 0.75 | 0.362 | 0.049 | 0.220 | n |
| 0.25 < EDS ≤ 0.50 | 0.284 | 0.078 | 0.351 | $\epsilon$ |
| 0.00 < EDS ≤ 0.25 | 0.203 | 0.108 | 0.485 | $\epsilon$ |
| EDS ≤ 0.00 | 0.098 | 0.373 | 1.206 | $\epsilon$ |

While the preceding analyses examine the theoretical geometry and sensitivity structure of the EDS, it is also instructive to assess how the score behaves in practical retrieval scenarios. We therefore conducted a structured perturbation analysis across 25 retrieval instances (two algorithms applied to four datasets and multiple variables). $\epsilon$, $\beta$ and $n$ were independently perturbed by ±5%, ±10%, ±20%, and ±30%, while keeping the remaining components unchanged. The results show that EDS responded smoothly to increasing perturbation magnitude, with the largest sensitivity associated with $\epsilon$, followed by $\beta$, and smaller effects for $n$. For example, a ±20% perturbation yields median absolute EDS changes of approximately 0.054 for $\epsilon$, 0.016 for $\beta$, and 0.014 for $n$. The comparatively smaller influence of $n$ reflects the fact that, for most practical retrieval instances, $n$ is close to its ideal value and contributes little to the total distance. This analysis demonstrates that, in real applications, the sensitivity of the EDS strongly depends on the relative contribution of each component for a given retrieval. For context, the relative contribution of the metrics for the same 25 retrieval instances are shown in Figure 3. The score is primarily driven by $\epsilon$, systematic bias ($\beta$) generally plays a secondary role, and contribution of the valid retrieval ratio ($n$) is small for most retrievals.

*Table 2. Sensitivity of the Euclidean Distance Score (EDS) to metric-level perturbations. Minimum, median, mean, and maximum absolute changes in EDS (| ΔEDS |) resulting from ±5%, ±10%, ±20%, and ±30% perturbations*

| Metric | Perturbation | Min \| ΔEDS \| | Median \| ΔEDS \| | Mean \| ΔEDS \| | Max \| ΔEDS \| |
|---|---|---|---|---|---|
| $\beta$ | 5% | 3.38E-06 | 3.89E-03 | 8.05E-03 | 6.10E-02 |
| $\beta$ | 10% | 6.58E-06 | 7.78E-03 | 1.61E-02 | 1.23E-01 |
| $\beta$ | 20% | 1.25E-05 | 1.55E-02 | 3.21E-02 | 2.52E-01 |
| $\beta$ | 30% | 1.77E-05 | 2.29E-02 | 4.78E-02 | 3.85E-01 |
| $\epsilon$ | 5% | 1.18E-03 | 1.38E-02 | 1.83E-02 | 6.10E-02 |
| $\epsilon$ | 10% | 2.34E-03 | 2.75E-02 | 3.66E-02 | 1.23E-01 |
| $\epsilon$ | 20% | 4.59E-03 | 5.38E-02 | 7.31E-02 | 2.52E-01 |
| $\epsilon$ | 30% | 6.73E-03 | 7.87E-02 | 1.09E-01 | 3.85E-01 |
| $n$ | 5% | 0.00E+00 | 2.56E-03 | 4.36E-03 | 2.65E-02 |
| $n$ | 10% | 0.00E+00 | 8.07E-03 | 1.18E-02 | 7.16E-02 |
| $n$ | 20% | 0.00E+00 | 1.44E-02 | 3.33E-02 | 1.69E-01 |
| $n$ | 30% | 0.00E+00 | 2.28E-02 | 6.10E-02 | 2.68E-01 |

[Figure]

*Figure 3. Fractional contribution of the EDS components to the squared distance for each retrieval instance (stacked bars), with the corresponding EDS shown on the secondary axis. Contributions are shown for $\epsilon$, $\beta$, and $n$.*

**B.  Misinterpretation of Reduced Major Axis (RMA) Regression**

We thank the reviewer for this comment. In the revised EDS framework, regression slope and the Pearson correlation coefficient are no longer included in the score due to identified issues such as the statistical dependency between metrics (see Reply D for more details on redundancy). As a result, the revised EDS no longer treats slope and correlation as independent dimensions, thereby addressing the concern raised by the reviewer.

**C.  Ideal Point**

We thank the reviewer for this comment. The ideal point in the EDS is not intended to represent a physically attainable retrieval result, but rather a reference used to define a distance-based measure of relative performance. In the revised formulation, the ideal point is defined solely in terms of zero error, zero bias, and full retrieval success, which serve as consistent reference values for quantifying deviations among algorithms, not as targets expected to be reached in practice.

**D.  Redundancy**

We thank the reviewer for raising the concern about metric redundancy. We agree that jointly retaining metrics that capture the same performance aspect can lead to redundancy and violate the assumptions underlying Euclidean aggregation. This concern directly motivated the revision of the EDS metric set.

To explicitly assess redundancy among candidate metrics, we analysed pairwise inter-metric relationships considering 25 retrieval instances (two algorithms applied to four datasets and different variables, including $a_{ph}(\lambda)$, $b_{bp}(\lambda)$, , $a_{dg}(\lambda)$, Chla, SPM, $kd(\lambda)$ and Secchi depth). Inter-metric relationships were quantified using Spearman's rank correlation coefficient (Figure 4), which is appropriate for assessing monotonic associations in small samples. Pairwise scatterplots of the raw metric values provide a visual check of the corresponding relationships.

This analysis indicates that several candidate metrics exhibit substantial dependence and should not be jointly retained in a distance-based score. In particular, regression slope and the Pearson correlation coefficient ($r$) show systematic associations with error magnitude- and bias-based metrics respectively and were therefore considered redundant. A complementary rank-based variance inflation factor (VIF) analysis further supports this conclusion: while the original formulation exhibited elevated VIF values (up to ~2.7 for $\varepsilon$ and ~2.6 for r), indicating shared variance among metrics, all retained components in the revised EDS exhibit consistently low VIF values (≤ ~1.8), confirming minimal multicollinearity.

Based on these results, regression slope and correlation were removed from the EDS, and only one representative metric of error magnitude ($\varepsilon$), one of systematic bias ($\beta$), and one of retrieval robustness (n) were retained. The revised EDS therefore avoids double-counting of performance aspects and satisfies the requirement that each dimension captures complementary information, addressing the redundancy concern raised by the reviewer. Detailed correlation and redundancy analyses will be provided in a revised manuscript.

[Figure]

*Figure 4. Pairwise Spearman rank correlation coefficients (ρ) between candidate metrics evaluated across 25 model–dataset–variable instances. The lower triangular matrix shows correlation coefficients with statistical significance indicated by asterisks, while the upper triangle displays scatterplots of the corresponding metric pairs for illustrative purposes.*

**E.  Behaviour in the Example Cases**

We thank the reviewer for this comment. We agree that the examples highlighted in the manuscript revealed limitations of the original EDS formulation, particularly the disproportionate influence of regression-based metrics on the final score.

In the $b_{bp}$ retrieval case, the originally low EDS value was indeed driven primarily by a very high regression slope (which had high uncertainty in its estimation due to small range), despite low bias ($\beta \approx -4\%$) and acceptable median symmetric accuracy ($\epsilon \approx 24\%$). In the revised formulation, regression slope and correlation are no longer included in the distance calculation. As a result, the revised EDS for this case increases to 0.75, which is consistent with the overall assessment indicated by the error, bias, and robustness metrics.

In the oligotrophic $k_d$ example, the original formulation yielded a moderate EDS despite low error and bias, due to the inclusion of correlation-based diagnostics. With the revised EDS, the score increases to 0.86. This value is consistent with the corresponding EDS obtained when considering all trophic states (0.82) and when analysing trophic regimes separately (mesotrophic: 0.84; eutrophic: 0.78), indicating coherent behaviour with stratifications and the change of dynamic range. Because agreement-based metrics remain comparable across these stratifications, consistency among the performance is expected. Such consistency was not observed in the original formulation, where association-based metrics introduced sensitivity to changes in data range.

In the revised manuscript, these updated EDS values will replace the original scores in the example cases, and we will additionally discuss the behaviour of commonly used metrics, including regression slope, highlighting situations in which they may yield misleading conclusions.

**F.   Treatment of Variables**

We thank the reviewer for this comment. The EDS aims to provide a unified, dimensionless framework for summarizing retrieval performance. A key design feature of the revised EDS is that all retained components are expressed in relative or fractional terms that are range-independent: error magnitude and bias are quantified a percentage deviations, and retrieval robustness is expressed as a fraction. The regression slope and Pearson correlation coefficient, present in the original formulation, are indeed more sensitive to factors such as data dynamic range, which can compromise cross-variable comparability. As such, with the revised version we believe cross-variable comparisons are plausible.

---

## Author Comment (AC4)

**Authors Reply to RC4: 'Comment on egusphere-2025-4343', Anonymous Referee #4, 09 Dec 2025**

*Reviewer comments appear in **bold**. Authors' responses are in plain text.

**Using a single metric to assess the performance of bio-optical algorithms is an interesting but challenging topic. The Euclidean Distance Score (EDS) proposed by the authors could be a useful approach. My major concerns, similar to those raised by other reviewers, involve the choice of individual metrics, their robustness, existing correlations, and inconsistent ranges (i.e., lack of normalization). These issues are critical and should be properly addressed.**

**From a potential user's perspective, the selection of metrics is somewhat arbitrary, and the results can therefore be misleading. Although the mathematics behind the Euclidean distance is straightforward, the method implicitly assumes that all metrics are equally important, which is not always the case and may vary across application scenarios. For example, in water-quality monitoring, absolute error may be the most important metric, while in time-series studies, bias, which indicates systematic over- or underestimation, may be more relevant. A combined score may hide such differences. This raises the question of how sensitive the EDS is to each input metric. The authors may consider performing a sensitivity analysis to examine whether all selected metrics contribute equally or whether some dominate the score.**

**Overall, the paper is well written. I was pleased to read it. However, the issues above need to be properly addressed for the paper to contribute to the field.**

We thank the reviewer for this thoughtful assessment and for raising concerns that are indeed central to the design of a composite performance metric. These considerations, together with related comments from other reviewers, directly motivated revisions and further development of the EDS framework.

Regarding the perceived arbitrariness in metric selection, the revised manuscript will include a dedicated section detailing the selection process. This includes a characterization of candidate metrics (summarized in Table 3) and a redundancy analysis (more details on Reply to Reviewer 3, D) to ensure that only complementary metrics are retained. Based on this analysis, the EDS was restricted to three dimensionless metrics representing distinct performance aspects: error magnitude (median symmetric accuracy, $\epsilon$), systematic bias (symmetric signed percentage bias, $\beta$), and retrieval robustness (valid retrieval ratio, $n$). Metrics found to be redundant, range-dependent, or sensitive to regression assumptions (e.g., slope and correlation) were removed.

Figure 1 illustrates a geometric representation of the revised EDS in a three-dimensional space, following the exclusion of regression slope and Pearson correlation coefficient from the score . Of the three remaining metrics, the valid retrieval ratio ($n$) is naturally bounded between 0 and 1 and referenced to an ideal value of unity. Median Symmetric Accuracy ($\epsilon$) and Symmetric Signed Percentage Bias ($\beta$) are dimensionless, defined relative to an ideal value of zero, with magnitudes that directly reflect fractional deviations from perfect agreement. Although they are formally unbounded and may exceed unity, we choose to not normalize them so that extreme deviations are strongly penalized rather than compressed through imposed bounds. For retrievals with errors and biases below 100% (corresponding to reasonably performing retrievals in practice), all metrics are of

order unity and therefore can contribute comparably to the distance. This is reflected in the approximately isotropic geometry of the high-EDS region, indicating that no single metric is implicitly favoured in that regime. In contrast, values of $\epsilon$ or $|\beta|$ exceeding unity correspond to strongly degraded retrievals and dominate the distance, displacing the solution away from the ideal point in the EDS space.

[Figure]

*Figure 1. Geometric representation of the Euclidean Distance Score (EDS) in the three-dimensional $(\beta, \epsilon, n)$ space. The ideal retrieval corresponds to $(\beta, \epsilon, n) = (0,0,1)$. The shown domain is restricted to metric combinations satisfying $| \beta | \leq \epsilon$, consistent with their definition. For visualization purposes, EDS values are displayed over the range $[-2,1]$.*

Nevertheless, having comparable numerical scales does not imply that all metrics exert equal influence on the EDS across its admissible domain. The components exhibit different empirical variances: the agreement-based terms ($\epsilon$ and $\beta$) may span a wide range depending on retrieval quality, whereas the valid retrieval ratio ($n$) is bounded and, in most realistic applications, concentrated near its ideal value of unity. To quantify how these differences translate into effective influence on the score, we performed a pointwise sensitivity analysis based on the analytical gradient of the Euclidean distance, identifying the locally dominant direction of score variation at each admissible point.

The resulting dominance structure, illustrated in Figure 2, shows that variations in error magnitude ($\epsilon$) control the sensitivity of the score over most of the admissible space. Systematic bias ($\beta$) does not emerge as a dominant sensitivity on its own, but attains equal influence with $\epsilon$ along a narrow, well-defined surface where $|\beta| = \epsilon$ and both exceed $n - 1$. Sensitivity to the valid retrieval ratio ($n$) is comparatively smaller over large portions of the space, but becomes dominant where $\epsilon$ and $\beta$ are low.

When averaged over the explored domain (restricted to EDS > −2), the mean relative sensitivities are 1.106 for $\epsilon$, 0.338 for $\beta$, and 0.116 for $n$. These values describe the average local responsiveness of the EDS to perturbations in each component across the admissible space. Stratifying the analysis by EDS (Table 1) reveals a transition in sensitivity regimes: near-optimal retrievals (EDS > 0.75) are most sensitive to $n$, whereas increasingly degraded retrievals exhibit progressively stronger sensitivity to $\epsilon$. The comparatively lower sensitivity associated with $\beta$ does not imply a negligible contribution of bias to the distance. Rather, it reflects the constraint $|\beta| \leq \epsilon$, which limits the independent variability of bias and confines its influence on the distance to specific regions of the space where systematic over- or underestimation occurs.

[Figure]

Figure 2. Sensitivity dominance structure of the Euclidean Distance Score (EDS) in the three-dimensional ($\beta, \epsilon, n$) space under the constraint $|\beta| \leq \epsilon$. Colored regions indicate the metric to which the EDS is locally most sensitive, based on the analytical gradient of the distance

*Table 1. Mean relative sensitivities of the Euclidean Distance Score (EDS) with respect to error magnitude (ε), systematic bias (β), and valid retrieval ratio (n), computed over non-overlapping EDS bins. Sensitivities quantify the average local response of the score to perturbations in each component.*

| EDS range | ⟨Sn⟩ | ⟨Sβ⟩ | ⟨Sε⟩ | Dominant sensitivity |
|---|---|---|---|---|
| EDS > 0.75 | 0.425 | 0.023 | 0.103 | n |
| 0.50 < EDS ≤ 0.75 | 0.362 | 0.049 | 0.220 | n |
| 0.25 < EDS ≤ 0.50 | 0.284 | 0.078 | 0.351 | ε |
| 0.00 < EDS ≤ 0.25 | 0.203 | 0.108 | 0.485 | ε |
| EDS ≤ 0.00 | 0.098 | 0.373 | 1.206 | ε |

While the preceding analyses examine the theoretical geometry and sensitivity structure of the EDS, it is also instructive to assess how the score behaves in practical retrieval scenarios. We therefore conducted a structured perturbation analysis across 25 retrieval instances (two algorithms applied to four datasets and multiple variables). $\epsilon$, $\beta$ and $n$ were independently perturbed by ±5%, ±10%, ±20%, and ±30%, while keeping the remaining components unchanged. The results show that EDS responded smoothly to increasing perturbation magnitude, with the largest sensitivity associated with $\epsilon$, followed by $\beta$, and smaller effects for $n$. For example, a ±20% perturbation yields median absolute EDS changes of approximately 0.054 for $\epsilon$, 0.016 for $\beta$, and 0.014 for $n$. The comparatively smaller influence of $n$ reflects the fact that, for most practical retrieval instances, $n$ is close to its ideal value and contributes little to the total distance. This analysis demonstrates that, in real applications, the sensitivity of the EDS strongly depends on the relative contribution of each component for a given retrieval. For context, the relative contribution of the metrics for the same 25 retrieval instances are shown in **Error! Reference source not found.**. The score is primarily driven by $\epsilon$, systematic bias ($\beta$) generally plays a secondary role, and contribution of the valid retrieval ratio ($n$) is small for most retrievals.

*Table 2. Sensitivity of the Euclidean Distance Score (EDS) to metric-level perturbations. Minimum, median, mean, and maximum absolute changes in EDS (| ΔEDS |) resulting from ±5%, ±10%, ±20%, and ±30% perturbations*

| Metric | Perturbation | Min \| ΔEDS \| | Median \| ΔEDS \| | Mean \| ΔEDS \| | Max \| ΔEDS \| |
|---|---|---|---|---|---|
| $\beta$ | 5% | 3.38E-06 | 3.89E-03 | 8.05E-03 | 6.10E-02 |
| $\beta$ | 10% | 6.58E-06 | 7.78E-03 | 1.61E-02 | 1.23E-01 |
| $\beta$ | 20% | 1.25E-05 | 1.55E-02 | 3.21E-02 | 2.52E-01 |
| $\beta$ | 30% | 1.77E-05 | 2.29E-02 | 4.78E-02 | 3.85E-01 |
| $\epsilon$ | 5% | 1.18E-03 | 1.38E-02 | 1.83E-02 | 6.10E-02 |
| $\epsilon$ | 10% | 2.34E-03 | 2.75E-02 | 3.66E-02 | 1.23E-01 |
| $\epsilon$ | 20% | 4.59E-03 | 5.38E-02 | 7.31E-02 | 2.52E-01 |
| $\epsilon$ | 30% | 6.73E-03 | 7.87E-02 | 1.09E-01 | 3.85E-01 |
| $n$ | 5% | 0.00E+00 | 2.56E-03 | 4.36E-03 | 2.65E-02 |
| $n$ | 10% | 0.00E+00 | 8.07E-03 | 1.18E-02 | 7.16E-02 |
| $n$ | 20% | 0.00E+00 | 1.44E-02 | 3.33E-02 | 1.69E-01 |
| $n$ | 30% | 0.00E+00 | 2.28E-02 | 6.10E-02 | 2.68E-01 |

[Figure]

*Figure 3. Fractional contribution of the EDS components to the squared distance for each retrieval instance (stacked bars), with the corresponding EDS shown on the secondary axis. Contributions are shown for $\epsilon$, $\beta$, and $n$.*

We also acknowledge that different applications may prioritize different performance aspects (e.g., error magnitude versus bias), thank you for raising this point. The EDS was conceptualized for typical algorithm validation in aquatic remote sensing. We do however highlight that the EDS formulation could be adapted to give more weight to different aspects according to the needs of the user. In the revised manuscript we will highlight this more explicitly.

*Table 3. Summary of candidate performance metrics considered for the evaluation of bio-optical retrieval algorithms. For each metric, the table reports its mathematical definition, metric class, key characteristics, and main limitations when applied to bio-optical variables. In all definitions, $E_i$ and $O_i$ denote the estimated and observed values of the i-th retrieval, respectively, and $Q_i = E_i/O_i$ is the corresponding accuracy ratio.*

| Metric | Definition | Metric Class | Key characteristics | Limitations for bio-optical variables |
|---|---|---|---|---|
| **MAPE** | $$\frac{100}{n}\sum_{i=1}^{n}\left|\frac{E_i - O_i}{O_i}\right|$$ | Agreement (multiplicative, deviation-based) | - Average deviation in percentage format

 - Range-independent interpretability | - Unstable for small observed values

 - Asymmetric (overestimation penalized more than underestimation) |
| **RMSE** | $$\sqrt{\frac{1}{n}\sum_{i=1}^{n}(E_i - O_i)^2}$$ | Agreement (additive, deviation-based) | - Deviation in original units

 - Quadratic penalty emphasizes large errors | - Highly sensitive to outliers and heteroscedasticity

 - Interpretability affected by the range of the data |
| **MAE** | $$\frac{1}{n}\sum_{i=1}^{n}|E_i - O_i|$$ | Agreement (additive, deviation-based) | - Average deviation in original units

 - Less sensitive to outliers than RMSE | - Sensitive to heteroscedasticity

 - Interpretability affected by the range of the data |
| **Bias** | $$\frac{1}{n}\sum_{i=1}^{n}(E_i - O_i)$$ | Agreement (additive, signed deviation-based) | - Average signed deviation in original units

 - Measure of systematic over- /under estimation | - Sensitive to heteroscedasticity

 - Interpretability affected by the range of the data |
| **MAE-ratio** | $$10^{\frac{1}{n}\sum_{i=1}^{n}|\log_{10}(Q_i)|}$$ | Agreement (multiplicative, ratio-based) | - Average deviation in ratio-based format

 - Range-independent interpretability

 - Suitable for log-normally distributed variables | - Mean aggregation remains sensitive to outliers and heteroscedasticity that could remain even in log space |
| **Bias-ratio** | $$10^{\frac{1}{n}\sum_{i=1}^{n}\log_{10}(Q_i)}$$ | Agreement (multiplicative, signed ratio-based) | - Average signed deviation in ratio form

 - Measure of systematic over- /under estimation

 - Range-independent interpretability | - Mean aggregation remains sensitive to outliers and heteroscedasticity that could remain even in log space |

| Metric | Definition | Metric Class | Key characteristics | Limitations for bio-optical variables |
|---|---|---|---|---|
| | | | - Suitable for log-normally distributed variables | |
| **RMSE-ratio** | $10^{\frac{1}{n}\sum_{i=1}^{n}(\log_{10}(Q_i))^2}$ | Agreement (multiplicative, ratio-based) | - Deviation in a ratio format

- Quadratic penalty emphasizes large errors

- Range-independent interpretability

- Suitable for log-normally distributed variables | - Mean aggregation remains sensitive to outliers and residual heteroscedasticity that could remain even in log space |
| **Median Symmetric Accuracy (ϵ)** | $10^{median(|\log_{10}(Q_i)|)} - 1$ | Agreement (multiplicative, deviation-based) | - Median proportional deviation

- Range-independent interpretability

- Suitable for log-normally distributed variables

- Median aggregation yields robustness to outliers and residual heteroscedasticity | - Less familiar metric |
| **Symmetric Signed Percentage Bias (β)** | $\text{sign}(M)(10^{|M|} - 1)$

where $M = \text{median}(\log_{10}(Q_i))$ | Agreement (multiplicative, signed deviation-based) | - Median signed proportional deviation (systematic bias)

- Range-independent interpretability

- Suitable for log-normally distributed variables

- Median aggregation yields robustness to outliers and residual heteroscedasticity | - Less familiar metric |
| **Pearson correlation coefficient (r)** | $r = \dfrac{cov(E,O)}{\sigma_E \sigma_O}$ | Association | - Strength of linear association (typically in log space) | - Does not quantify agreement

- Lacks response to bias |

| Metric | Definition | Metric Class | Key characteristics | Limitations for bio-optical variables |
|---|---|---|---|---|
| | | | | - Sensitive to outliers, leverage points and data range |
| **Regression slope** | Regression method dependent | Association | - Describes how variability in the observations is scaled by the estimates (typically in log space) | - Does not quantify agreement

- Sensitive to outliers and leverage points

- Poorly conditioned for narrow data ranges |
| **Intercept** | Regression method dependent | Association | - Describes the offset between estimated and observed values at the origin (typically in log space) | - Limited physical interpretability |
| **Valid retrieval ratio (n)** | $\dfrac{N_{E,valid}}{N_O}$ | Algorithm robustness | - Fraction of cases for which a valid retrieval is produced

- Captures algorithm convergence and ability to produce physically plausible outputs | - Depends on the definition of the valid retrieval range, which can be subjective |